# SARS-CoV-2 detection in multi-sample pools in a real pandemic scenario: A screening strategy of choice for active surveillance

Andrés Marcos Castellaro[1,2☯]*, Pablo Velez[3☯], Guillermo Giaj Merlera[3], Juan Rondan Dueñas[3], Felix Condat[3], Jesica Gallardo[3], Aylen Makhoul[3], Camila Cinalli[3], Lorenzo Rosales Cavaglieri[3], Guadalupe Di Cola[4,5], Paola Sicilia[6], Laura López[7], Facultad de Ciencias Químicas UNC Group[1¶], José Luis Bocco[1,2], María Gabriela Barbás[8], Diego Hernán Cardozo[8], María Belén Pisano[4,5], Viviana Ré[4,5], Andrea Belaus[3], Gonzalo Castro[6]

1 Departamento de Bioquímica Clínica, Facultad de Ciencias Químicas, Universidad Nacional de Córdoba, Córdoba, Argentina, 2 Centro de Investigaciones en Bioquímica Clínica e Inmunología (CIBICI), CONICET, Universidad Nacional de Córdoba, Córdoba, Argentina, 3 Unidad de Biología Molecular, Centro de Excelencia en Productos y Procesos de Córdoba (CEPROCOR), Córdoba, Argentina, 4 Instituto de Virología "Doctor José María Vanella", Facultad de Ciencias Médicas, Universidad Nacional de Córdoba, Córdoba, Argentina, 5 Consejo Nacional de Investigaciones Científicas y Técnicas (CONICET), Córdoba, Argentina, 6 Laboratorio Central, Ministerio de Salud de la Provincia de Córdoba, Córdoba, Argentina, 7 Dirección de Epidemiología, Ministerio de Salud de la Provincia de Córdoba, Córdoba, Argentina, 8 Secretaría de Prevención y Promoción de la Salud, Ministerio de Salud de la Provincia de Córdoba, Córdoba, Argentina

☯ These authors contributed equally to this work.
¶ Membership of the Facultad de Ciencias Químicas UNC Group is provided in the Acknowledgments.
* andres.castellaro@unc.edu.ar

**Data Availability Statement:** All relevant data are within the paper.

**Funding:** This study was funded by the Ministry of Health of the Province of Córdoba, Argentina as

## Abstract

### Background

The current COVID-19 pandemic has overloaded the diagnostic capacity of laboratories by the gold standard method rRT-PCR. This disease has a high spread rate and almost a quarter of infected individuals never develop symptoms. In this scenario, active surveillance is crucial to stop the virus propagation.

### Methods

Between July 2020 and April 2021, 11,580 oropharyngeal swab samples collected in closed and semi-closed institutions were processed for SARS-CoV-2 detection in pools, implementing this strategy for the first time in Córdoba, Argentina. Five-sample pools were constituted before nucleic acid extraction and amplification by rRT-PCR. Comparative analysis of cycle threshold (Ct) values from positive pools and individual samples along with a cost-benefit report of the whole performance of the results was performed.

### Results

From 2,314 5-sample pools tested, 158 were classified as positive (6.8%), 2,024 as negative (87.5%), and 132 were categorized as indeterminate (5.7%). The Ct value shift due to sample dilution showed an increase in Ct of 2.6±1.53 cycles for N gene and 2.6±1.78 for

part of the program for active surveillance for SARS-CoV-2 dissemination and received material support from the National University of Córdoba. The funders had no role in study design, data collection and analysis, decision to publish, or preparation of the manuscript.

**Competing interests:** The authors have declared that no competing interests exist.

ORF1ab gene. Overall, 290 pools were disassembled and 1,450 swabs were analyzed individually. This strategy allowed correctly identifying 99.8% of the samples as positive (7.6%) or negative (92.2%), avoiding the execution of 7,806 rRT-PCR reactions which represents a cost saving of 67.5%.

## Conclusion

This study demonstrates the feasibility of pooling samples to increase the number of tests performed, helping to maximize molecular diagnostic resources and reducing the work overload of specialized personnel during active surveillance of the COVID-19 pandemic.

## Introduction

The emergence of coronavirus disease 2019 (COVID-19) caused by severe acute respiratory syndrome coronavirus 2 (SARS-CoV-2) led humanity to an unprecedented pandemic with severe health consequences for the population worldwide. Diagnostic tests are essential due to their ability to detect and provide answers for pandemic management [1, 2]. However, they may be hampered because of the high demand that overwhelmed the healthcare system and the limited supply of reagents required for the setup of these tests [3]. This aspect has been particularly worse in some geographic regions of the world, in low and middle-income countries as well [4]. Thus, the context of SARS-CoV-2 pandemics motivated the design of alternative diagnostic strategies.

SARS-CoV-2 infection is characterized by the great diversity of signs and symptoms affecting patients. The most frequently associated with COVID-19 are fever, dry cough and generalized weakness, though symptoms such as nausea, diarrhea, loss of smell, pharyngitis and enlarged tonsils have also been reported [5]. However, about a quarter of infected people never develop symptoms (asymptomatic) and about half of the infected individuals do not manifest any symptom at the testing time (presymptomatic) [6, 7]. Both groups are of great concern as they contribute to the spread of the virus because they are not aware that they are infected. These groups are undetectable through passive surveillance but require instead active surveillance strategy based on massive testing methods. Hence, the need to increase testing capacity entails developing alternative strategies to optimize resources, save time and reduce labor demand, thereby enhancing COVID-19 diagnosis which, in turn, is essential for evaluating the disease spread and for tracing the contacts of infected individuals [8, 9]. Testing pooled samples by real-time reverse-transcriptase-polymerase chain reaction (rRT-PCR) could be a plausible way to deal with the huge demand for SARS-CoV-2 detection as long as it demonstrates optimum diagnostic performance. This strategy consists of mixing samples in a pool and then performing a single RNA extraction followed by a rRT-PCR assay. If the result is positive, then it must be identified which of the individual samples that make up the pool are positive by performing the RNA extraction and rRT-PCR test for each one of them. On the other side, if the rRT-PCR test is negative it is assumed that all individual samples composing the pool are also negative. Thus, according to this layout, it is expected that fewer nucleic acid extraction and rRT-PCR tests will be required, saving reagents, time and labor demand compared with analyzing individual samples. This pooling approach was previously developed for the analysis of several infectious diseases, e.g., malaria and HIV [10, 11], and is currently used as a screening method in blood banks prior to transfusion [12, 13]. Pooling nasopharyngeal swab samples for RNA virus detection, such as influenza has already been evaluated [14]. In the case of COVID-

19, the method was considered and analyzed as strategy by countries like Germany, Israel and the United States [15–17], after the Food and Drug Administration (FDA) approved the emergency use of Quest SARS-CoV-2 rRT-PCR test for pooled samples on July, 2020 [18].

The aim of this work was the implementation of a multi-sample pool strategy during a specific period of the COVID-19 pandemic, to decrease costs and response times, while increasing efficiency and testing capacity, thereby contributing to early patient assistance and control of disease spread.

## Methods

### Sample pooling

In July 2020, when the background SARS-CoV-2 community prevalence was between 3% and 5% in Córdoba, Argentina [19], oropharyngeal swab samples in viral transport media (Jun Nuo, Chengwu County, Shandong Province, China) obtained by healthcare personnel in testing centers, were processed using a pooling strategy, following the protocol described by Ambrosi et al. with minor modifications [20]. Briefly, from five individual oropharyngeal swab samples, an aliquot of 60 μL of each was taken to create a pool, with a final volume of 300 μL. Each pool of samples was processed for RNA extraction and subsequent rRT-PCR analysis. From August 2020 to April 2021, when the community viral prevalence exceeded 5%, oropharyngeal swab samples were collected in 234 closed and semi-closed institutions. Each group of 5 sequentially obtained samples was pooled, without mixing samples from different institutions. Overall, 2,314 pools were made up, each one containing a mix of 5 individual samples, from initially 11,580 patient samples.

### Nucleic acid extraction and SARS-CoV-2 detection

RNA extraction of the pool was performed using Bioer MagaBio plus Virus DNA/RNA Purification Kit in addition to Bioer GenePure Pro fully automatic Nucleic Acid purification System (Bioer, Hangzhou, China) and EasyPure Viral DNA/RNA Kit (TransGene, Beijing, China), according to the manufacturer's instructions. A multiplex single step real time RT-PCR was carried out for amplification, using DisCoVery SARS-CoV-2 rRT-PCR Detection Kit (TransGen Biotech Co., Ltd, Beijing, China), which is designed to detect two SARS-CoV-2 target genes: Open Reading Frame 1ab (ORF1ab) and nucleocapsid (N), along with the human ribonuclease P (RNAseP) gene as endogenous control (Safecare Biotech Hangzhou, China).

### Classification criteria

Considering the cycle threshold (Ct) value for ORF1ab and N genes, the pools were classified as:

- Positive: pools with amplification of both genes with Ct≤38.

- Negative: those without amplification of both genes, or with fluorescence signal only in one of them with Ct>40.

- Suspected positive or indeterminate: pools that showed amplification in only one gene with Ct≤40 and those that amplified both genes with Ct>38.

To validate the previous results, all tests must amplify the endogenous RNAseP control gene.

## Data processing

Results obtained by rRT-PCR (sample code and their respective Ct) were organized and converted into comma-delimited files. These data were incorporated into a database based on APACHE, MYSQL and PHPMYADMIN. A web interface was created to monitor the tests, evaluate coherence through PHP scripts, visualize the data pools and their disassembly, and carry out searches.

## Statistical analysis

Differences in Ct values, mean, standard deviation and the reduction in the number of tests were calculated using R version 4.0.5, 2021. The delta Ct value (ΔCt) was defined as the absolute change in Ct value when the pooled sample was tested compared to the positive sample that composed the pool, when it was tested individually. Therefore, a positive ΔCt value (i.e., an increase in Ct value of the pooled sample) represents the loss of rRT-PCR sensitivity as a consequence of individual sample dilution within a pool composed of 5 samples [21]. ANOVA tests were done to compare groups, and a *P*-value of 0.05 or less was considered statistically significant.

## Ethics statement

Ethical review, approval and written informed consent from the participants were not required for the study on oropharyngeal swab samples obtained from human participants in accordance with the local legislation and institutional requirements. The Government of the Province of Córdoba waives the ethical review of the SARS-CoV-2 detection in multi-sample pools during 2020 and 2021 under strict confidentiality rules, based on the need for rapid surveillance and availability of methodologies for diagnosis of COVID-19.

## Results

The study included 11,570 samples analyzed under 2,314 pools format, containing 5 samples each. The results showed 158 positive (6.8%) and 2,024 negative pools (87.5%); also 132 pools were classified as indeterminate (5.7%). In total, 290 pools were disassembled (158 classified as positive and 132 suspected to be positive) to analyze each one of the 1,450 samples that composed them. As a result, the pooling strategy saved 7,806 tests, that is, 67.5% fewer tests were required for the screening, leading to a two-thirds reduction in costs.

A hundred and five (105) of the positive pools contained a single positive sample, and 53 contained more than one positive sample. Results of Ct variation values comparing each pool and individual samples are shown in **Fig 1.**

It was observed that 5-sample pools containing one positive and four negative individual swabs samples, yielded higher Ct values than individual sample testing exceeded by 2.6 cycles on average for both, N and ORF1ab genes (2.6±1.53 cycles for N gene and, 2.6±1.78 for ORF1ab gene). The differences between the Ct value of pooled and individual samples (ΔCt) are illustrated in **Fig 2**, showing the same mean value and comparable variability for both target genes.

For most pools composed of more than one positive individual sample, the Ct values obtained were closer to the average of the individual Ct values, showing a pattern distribution in the middle area of the plot, and decreasing as the number of positive individual swabs increases within the same pool (**Fig 3**).

Regarding the 132 indeterminate pools, 19 contained only one positive sample (**Table 1**), 108 were constituted by all negative individual samples and five contained one suspected

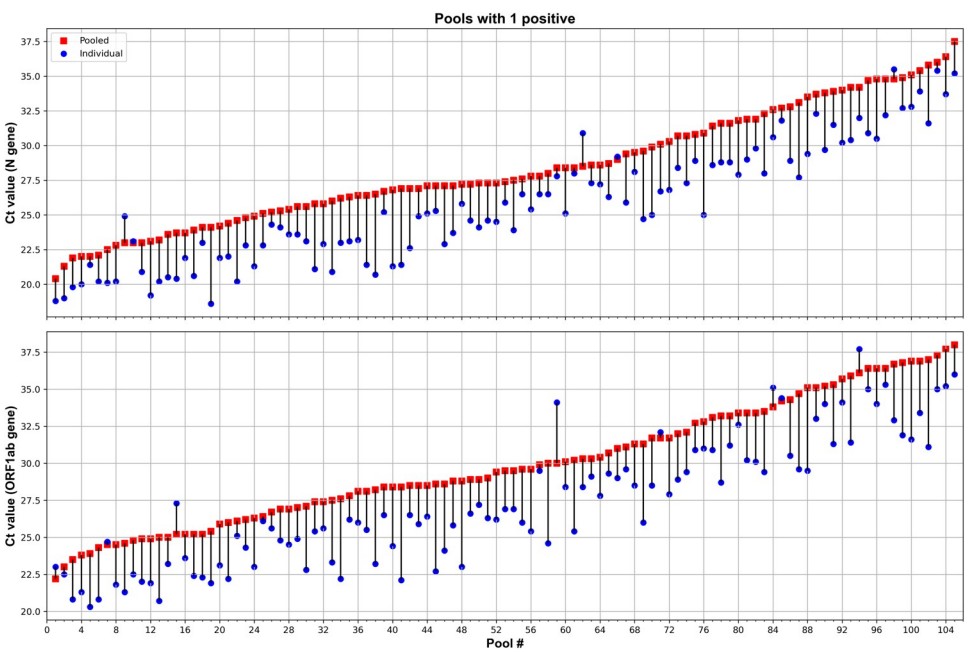

**Fig 1. Pools with a single positive.** Scatter plots showing cycle threshold (Ct) values and Ct shifts detected by rRT-PCR in five-samples pools (squares), composed of four negative and one positive sample, with respect to the individual positive samples (circles) for N (top graph) and ORF1ab (bottom graph) genes.

positive sample (inconclusive result according to the criteria provided in the user manual of the rRT-PCR kit, so a new sample was requested). Analysis of the Ct values distribution of the indeterminate group, indicates that there are no significant differences ($P<0.05$) for both, N and ORF1ab genes, regardless of whether any pool contained positive samples (**Fig 4**).

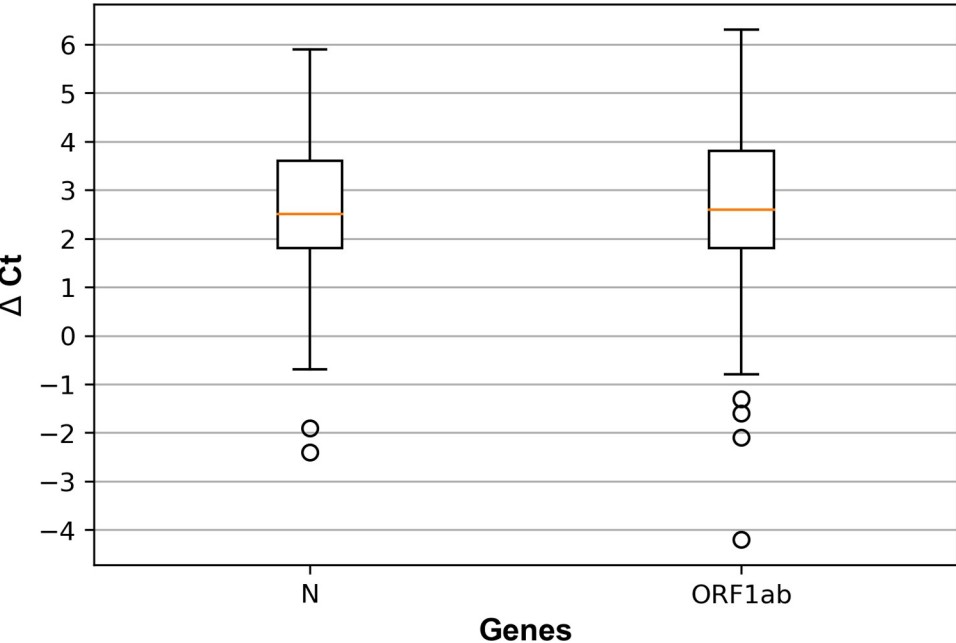

**Fig 2. Ct value shift due to sample pooling.** Box plot showing the difference between the Ct values (ΔCt) of five-samples pools (composed of 1 positive and 4 negative samples) and the respective individual positive samples for N and ORF1ab target genes.

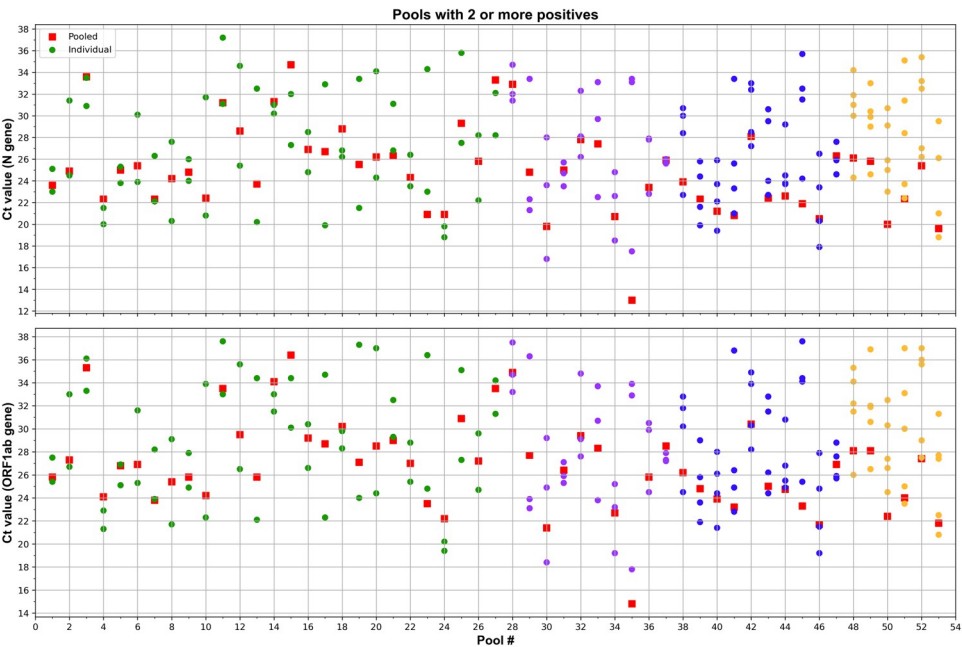

**Fig 3. Pools with two or more positives.** Scatter plots showing Ct values detected by rRT-PCR in five-samples pools (squares) and the individual positive samples (circles) from pools containing 2 (Pool # 1–27), 3 (Pool # 28–37), 4 (Pool # 38–47) or 5 (Pool # 39–53) positive samples. The analysis was performed for N (top graph) and ORF1ab (bottom graph) genes.

**Table 1. Ct values of positive individual samples from 19 indeterminate pools.**

| N gene | | ORF1ab gene | |
|---|---|---|---|
| **Pooled (Ct)** | **Individual Sample (Ct)** | **Pooled (Ct)** | **Individual Sample (Ct)** |
| 37.4 | 33.7 | NA | 36.1 |
| 35.7 | 34.6 | NA | 36.6 |
| NA | 37.9 | 38.3 | 36.7 |
| 32.9 | 29.6 | 39.9 | 31.7 |
| 35.2 | 33.0 | 41.0 | 35.0 |
| 37.0 | 33.5 | NA | 36.6 |
| 38.9 | 34.4 | NA | 36.5 |
| 36.3 | 32.7 | NA | 34.7 |
| 35.5 | 31.2 | NA | 36.2 |
| 37.3 | 32.1 | NA | 36.0 |
| 36.9 | 34.7 | NA | 37.7 |
| 34.6 | 31.1 | 38.4 | 32.9 |
| NA | 36.0 | 39.5 | 37.8 |
| 37.2 | 35.4 | NA | 37.5 |
| 39.9 | 36.4 | NA | 37.0 |
| 38.0 | 31.9 | NA | 34.4 |
| 35.8 | 32.7 | NA | 35.3 |
| 38.6 | 31.0 | NA | 33.5 |
| NA | 36.5 | 39.5 | 35.3 |

NA: no amplification.

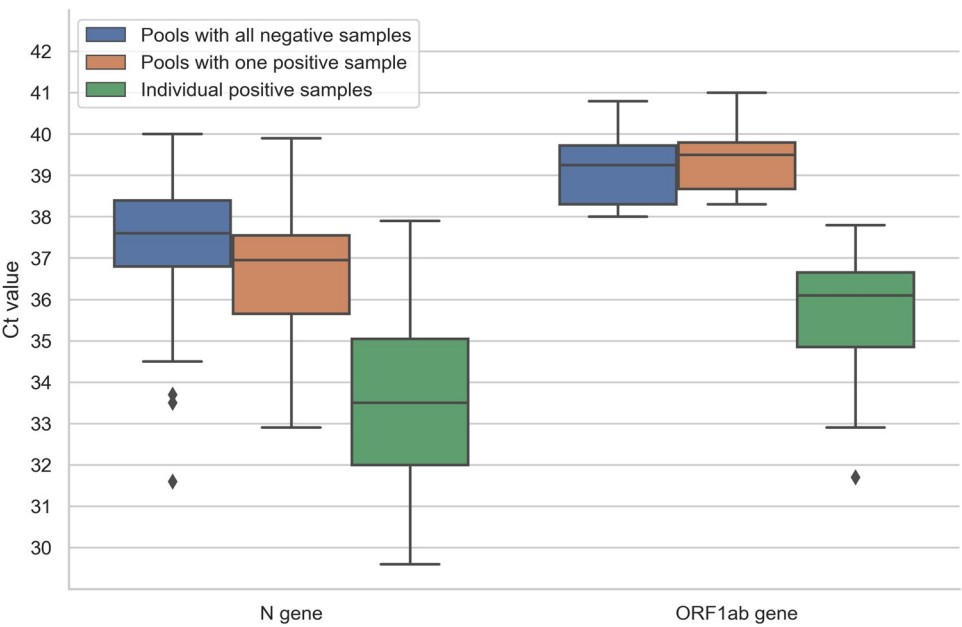

**Fig 4. Pools classified as indeterminate.** Box plot showing Ct values of N and ORF1ab genes from indeterminate pools which were finally classified as pools constituted by all negative samples (left), pools containing one positive sample (middle) and the individual positive samples after disassembly (right).

## Discussion

Given the high transmission rates of SARS-CoV-2 and that almost a quarter of infected people are asymptomatic [6, 7], one of the best strategies to control the virus spread is the early detection of those infected and their subsequent isolation. To achieve early detection of the infection, it is necessary to carry out active surveillance and massive testing. In this sense, different experimental or computational modeling tests have been carried out at the laboratory level to study the sensitivity and accuracy of the SARS-CoV-2 detection strategies using different grouping techniques. However, at present there are very few studies reporting results of some of these strategies applied in a real setting, demonstrating their applicability on a large scale with patient samples.

### Strengths of pooling strategy

This study demonstrates that the strategy based on the analysis of samples in the format of pools, consisting of 5 samples each, is a reliable approach for efficient detection of SARS-CoV-2 in a large number of samples in the context of the pandemic. In addition, it contributes to saving molecular diagnosis supplies and labor of the health personnel in areas or institutions with low viral circulation. It can be useful to monitor the infection rate in closed or semi-closed establishments such as residential homes, police and military headquarters, prisons or hospitals, due to these places being populated by those who are generally in close contact and/or at greater risk [20]. The emergence of the Omicron variant, which is more contagious than the previous ones SARS-CoV-2 strains [22], increases the importance of analyzing closed communities through periodic testing. For this, rRT-PCR testing using the pooling strategy has the advantage of being more specific and sensitive than rapid antigen tests, making it possible to detect false-negative and asymptomatic people without increasing costs as much, thus contributing to better management of pandemic in particular time periods.

About suspected positive results, almost every indeterminate pool was classified in that way because only the N gene showed amplification, which could be due to the increased sensitivity of this gene, in contrast with the ORF1ab gene [23]. Although in this indeterminate group was found only 19 pools (14.4%) that contained positive samples, the mean and range of the Ct values of both, N and ORF1ab genes, were similar to those found for the remaining indeterminate pools. Hence, it was not possible to establish a more rigorous cut-off Ct value to avoid disassembling groups composed of samples that will ultimately be all negative. Therefore, it was mandatory to disassemble all indeterminate groups to avoid false-negative results. Nevertheless, this approach allowed the correct identification of 99.8% of samples as positive (7.6%) or negative (92.2%), without the necessity to perform 7,806 tests, thus, saving 67.5% of costs and labor.

The pooling approach addresses a variety of difficulties associated with the pandemic context, most notably the limited availability of reagents, supplies, equipment, high labor demand of laboratory workers, as well as the high cost. However, the decision to implement this strategy must consider the total testing capacity and the disease prevalence in a specific geographic area.

The high sensitivity of rRT-PCR assays makes pool testing an efficient system that can be applied for resource optimization when the positivity rate is low (e.g., 5% or lower), improving laboratory testing capacities without additional requirements in terms of equipment availability or qualified personnel [24–26].

## Limitations of pooling strategy

It is important to highlight that optimal pool size must be determined according to the prevalence in the area under study [8, 27]. Previous studies have analyzed different pool sizes, from 2- to 64-samples [20, 25], and agreed that the number of samples in the pool is inversely proportional to the test sensitivity [26, 28]. It has been reported that SARS-CoV-2 prevalence is a suitable criterion to define the most efficient pool size, defining 5-sample groups as appropriate when the prevalence rates are about 5% [8, 20]. According to our findings, as well as previous reports, pooling samples entails the loss of sensitivity and an increase of the Ct value of pools compared with the individual specimen due to the sample dilution, especially for samples with low viral loads [26, 28, 29]. The increase in the Ct value observed in this study (2.6 cycles on average) for both target genes was comparable to those described in other reports of SARS-CoV-2 testing in pools constituted by 5 individual samples, where ΔCt ranges were between 2 and 3.4 [20, 28, 30, 31]. This variation results in more tests classified as indeterminate than would be obtained in a single sample approach, mainly in those groups that contained individual positive samples with a Ct value close to 38. In both positive and indeterminate pools, the Ct value distribution for the N and the ORF1ab genes shows the same trend observed in individual samples: lower Ct values were detected for the N gene than for the ORF1ab target sequence.

The limitation of this strategy to test the internal control of each sample, which is required to control the specimen quality, must also be underlined. So, false-negative results may occur if samples are improperly collected, transported or handled. Hence, negative results obtained by pooled sampling do not preclude SARS-CoV-2 infection and should not be used as the only criteria for treatment or for other social management decisions.

## Conclusion

The results obtained in this study show that the implementation of the pooling strategy was able to save 67.5% of rRT-PCR reactions in a low viral circulation scenario. Testing in pools

was a positive approach that expanded sample processing capabilities, allowing massive testing and early outbreaks detection. This experience could be taken into account as a strategy of active surveillance in hospitals, care homes, schools and other closed and semi-closed institutions. In a new post-pandemic scenario, with an expected decrease of viral circulation due to vaccine programs, the pooling approach could be implemented to carry out periodically large-scale testing to the population.

## Acknowledgments

The authors would like to acknowledge health care workers who take and processed the samples, all the staff of CEPROCOR laboratory, the CIBICI-CONICET research center team and the department of Bioquímica Clínica of Facultad de Ciencias Químicas, Universidad Nacional de Córdoba for their invaluable work and committed attitude in favor of the community throughout the SARS-CoV-2 pandemic.

Full list of members of the Facultad de Ciencias Químicas UNC Group: Pilar Crespo[1]; Gabriela Furlán[1]; Laura Gatica[1]; María Soledad Miró[1]; Rubén Motrich[1]; Ana Cristina Racca[1] and Luciana Reyna[1] (Lead author), email: luciana.reyna@unc.edu.ar.

## Author Contributions

**Conceptualization:** Andrés Marcos Castellaro, María Gabriela Barbás, María Belén Pisano, Viviana Ré, Gonzalo Castro.

**Data curation:** Andrés Marcos Castellaro, Pablo Velez, Guadalupe Di Cola, José Luis Bocco, Gonzalo Castro.

**Formal analysis:** Andrés Marcos Castellaro, Pablo Velez, Guadalupe Di Cola, José Luis Bocco, María Belén Pisano, Viviana Ré, Andrea Belaus.

**Funding acquisition:** María Gabriela Barbás, Diego Hernán Cardozo.

**Investigation:** Andrés Marcos Castellaro, Pablo Velez, Guillermo Giaj Merlera, Juan Rondan Dueñas, Felix Condat, Jesica Gallardo, Aylen Makhoul, Camila Cinalli, Lorenzo Rosales Cavaglieri, Guadalupe Di Cola, Paola Sicilia, Laura López, María Belén Pisano, Viviana Ré, Andrea Belaus, Gonzalo Castro.

**Methodology:** Andrés Marcos Castellaro, Pablo Velez, Guillermo Giaj Merlera, Juan Rondan Dueñas, Felix Condat, Jesica Gallardo, Aylen Makhoul, Camila Cinalli, Lorenzo Rosales Cavaglieri, Guadalupe Di Cola, Paola Sicilia, Laura López, María Belén Pisano, Viviana Ré, Andrea Belaus.

**Supervision:** Andrés Marcos Castellaro, José Luis Bocco, María Belén Pisano, Viviana Ré.

**Visualization:** Andrés Marcos Castellaro, María Belén Pisano, Viviana Ré.

**Writing – original draft:** Andrés Marcos Castellaro, Guadalupe Di Cola.

**Writing – review & editing:** Andrés Marcos Castellaro, Pablo Velez, Guadalupe Di Cola, José Luis Bocco, María Belén Pisano, Viviana Ré, Andrea Belaus.

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
