## [Decision Letter · Decision Letter 0]

17 Jan 2022

PONE-D-21-40099SARS-CoV-2 detection in multi-sample pools in a real pandemic scenario: a screening strategy of choice for active surveillancePLOS ONE

Dear Dr. Castellaro,

Thank you for submitting your manuscript to PLOS ONE. After careful consideration, we feel that it has merit but does not fully meet PLOS ONE’s publication criteria as it currently stands. Therefore, we invite you to submit a revised version of the manuscript that addresses the points raised during the review process.

ACADEMIC EDITOR: As appended below, the reviewers have raised  substantial concerns/critiques and suggested further justification/work to consolidate the findings. Do go through the comments and amend the MS accordingly. Afterwards, the MS should be checked by native speaker for grammar and syntax errors, if any. 

We look forward to receiving your revised manuscript.

Kind regards,

A. M. Abd El-Aty

Academic Editor

PLOS ONE

Journal Requirements:

"This study was part of the active surveillance for SARS-CoV-2 dissemination carried out by the Ministry of Health of the Province of Córdoba, Argentina. Authors received no specific grant from funding agencies in the public, commercial, or not-for-profit sectors. The funders had no role in study design, data collection and analysis, decision to publish, or preparation of the manuscript."

3. One of the noted authors is a group or consortium Facultad de Ciencias Químicas UNC Group. In addition to naming the author group, please list the individual authors and affiliations within this group in the acknowledgments section of your manuscript. Please also indicate clearly a lead author for this group along with a contact email address.

Reviewers' comments:

Reviewer's Responses to Questions

**Comments to the Author**

1. Is the manuscript technically sound, and do the data support the conclusions?

Reviewer #1: Yes

Reviewer #2: Yes

Reviewer #3: Yes

2. Has the statistical analysis been performed appropriately and rigorously? 

Reviewer #1: Yes

Reviewer #2: Yes

Reviewer #3: I Don't Know

3. Have the authors made all data underlying the findings in their manuscript fully available?

Reviewer #1: Yes

Reviewer #2: Yes

Reviewer #3: Yes

4. Is the manuscript presented in an intelligible fashion and written in standard English?

Reviewer #1: Yes

Reviewer #2: Yes

Reviewer #3: Yes

5. Review Comments to the Author

Reviewer #1: This is a generally well-written and informative manuscript. I have a few comments.

Abstract should explicitly state that 5 samples were used to create each pool.

Methods should present the sources of the VTM and the swabs. Even if this information is provided in a reference ,the methods section should state whether healthcare personnel obtained swabs or whether self-swabs were pooled, even if this information is provided in a reference.

Someplace the paper should tell us whether the pools represented truly random mixtures of cases, or whether perhaps multiple individuals from a single cluster of people (for example, family members or coworkers) were sometimes combined into a single pooled sample. The fact that 53 of 158 positive pools had more than one positive, while only 153 of 2314 pools contained any positives – a statistically improbable combination, suggests cluster effects or cross-contamination.

Reviewer #2: This study by Marcos Castallero et al., is very well-written, concise and timely, presenting a compelling argument for large-scale alternatives in testing strategies to cope with fluctuating test volumes. My only quibble is that the savings are limited to test reagents and laboratory resources and are partially mitigated by increased complexity in other parts of the testing chain, when 5% or less scenarios are transient and geographically variable, especially these days. For how long is pooling actually feasible and how does the lab switch seamlessly from pooling to not pooling? Or pooling for some samples but not for others. Can you make more of an argument about how pooling logistics fit into the overall strategy for surveillance and monitoring in low prevalence populations? Instead of just waiting for pooling to be useless, as it is right now? I realize this may be a bit out of your scope, but could add to your perspective on “the pandemic context”.

Omicron seems to have changed the game entirely. Although outside the timeframe in which your study was conducted, some recognition of how the pooling strategies you recommend are affected by these developments would be welcome. Any predictions about post-Omicrom scenarios – e.g. specific groups of people benefitting from a daily pooled sample prior to accessing a closed facility.

Data processing: I am surprised at all the scripts and databases required to handle two variables (testee identity and Ct values), but I have never worked with such a big sample set! Is there any chance of seeing some of the scripts used, database summaries and visualizations mentioned in this section?

Minor comments:

I think most journals would require a comma thousands separator.

line 95: “waives” is misspelled.

line 112: “Beijing” typo.

line 127-9: To be more concise, I would delete everything from “...exported” to “and interpreted with...”

line 139: Misplaced comma.

line 144: “All together” instead of “Altogether”

line 145-7: “saved” instead of “allowed to save”. Isn’t “67.5% fewer” the same thing as “3 times reduction” (“testing materials and costs were reduced by two thirds.”)?

line 173-4: “regardless of whether any pool contained positive samples.”

line 179: “disassembly” instead of “deconvolution” – stick to one word for testing all samples in a pool.

line 191: just “save” instead of “save costs of”.

line 195: “those” instead of “persons”, “at greater risk” instead of “risk factors”.

line 196: “optimal pool size” instead of “optimum size pool”

line 203: “especially for samples” instead of “affecting especially detection of samples”

line 212: “the increased sensitivity” instead “a more sensitivity”, “in contrast with the” instead of “contrasting with”

line 216: “disassembling” instead of “disarming”

line 219. Just “labor” instead of “decreasing personnel labor demanding”.

line 221: “deals” instead of “allows to deal”, “the pandemic context” instead of “a context of pandemics”

line 236: “expanded”, instead of “allow expanding” – replace period with comma after “capabilities”

Reviewer #3: This paper provides a comprehensive study of a pooled testing approach applied in a practical pandemic scenario. To my knowledge, there is not much data available regarding the trade-off between a potential loss of accuracy and the cost reduction resulting from the reduced number of tests. This paper provides such a comparison, giving an overview of the change of ct-values, the number of correct detections and the savings in terms of test reduction. This information is much needed to design public testing strategies based on pooled PCR tests., especially given the fact that antigen tests do not work well for the omicron variant and that there is still some hesitation with regards to the accuracy of pooled tests. For this reason, I find this study quite timely and recommend publication. The paper is also quite well-written, I do not have any specific suggestions or comments.

6. PLOS authors have the option to publish the peer review history of their article (what does this mean?). If published, this will include your full peer review and any attached files.

Reviewer #1: No

Reviewer #2: **Yes: **John Schellenberg

Reviewer #3: No

---

## [Author Response · Author response to Decision Letter 0]

12 Feb 2022

On behalf of my fellow authors, we wish to thank the three reviewers for their insightful, detailed and constructive criticisms of our manuscript. 

Responses to comments of reviewer #1

This is a generally well-written and informative manuscript. I have a few comments.

1-Abstract should explicitly state that 5 samples were used to create each pool.

Re.1: The abstract already had this data: On lines 39-40 the abstract states “Five-sample pools were constituted before nucleic acid extraction and amplification by rRT-PCR.”

2-Methods should present the sources of the VTM and the swabs. Even if this information is provided in a reference, the methods section should state whether healthcare personnel obtained swabs or whether self-swabs were pooled, even if this information is provided in a reference.

Re.2: We agree with the reviewer’s comment. To be clearer now at 'Sample pooling' section of Methods (lines 102-103) the following sentence has been included: "oropharyngeal swab samples in viral transport media (Jun Nuo, Chengwu County, Shandong Province, China), obtained by healthcare personnel in testing centers, were processed…"

3-Someplace the paper should tell us whether the pools represented truly random mixtures of cases, or whether perhaps multiple individuals from a single cluster of people (for example, family members or coworkers) were sometimes combined into a single pooled sample. The fact that 53 of 158 positive pools had more than one positive, while only 153 of 2314 pools contained any positives – a statistically improbable combination, suggests cluster effects or cross-contamination.

Re.3: We agree with the reviewer about this sharp right observation, indeed, it is very likely that there is a cluster effect since, as expressed in lines 107 and 108 of the manuscript ("From August 2020 to April 2021, when the community viral prevalence exceeded 5%, oropharyngeal swab samples were collected in 234 closed and semi-closed institutions."), many pools were formed with individual samples from closed and semi-closed institutions. To clarify on this point, the following sentence has been included: "These oropharyngeal swab samples were pooled together into 5 samples based on the order number, without mixing samples from different institutions." (lines 107-110).

Responses to comments of reviewer #2: 

1-This study by Marcos Castallero et al., is very well-written, concisate and timely, presenting a compelling argument for large-scale alternatives in testing strategies to cope with fluctuating test volumes. My only quibble is that the savings are limited to test reagents and laboratory resources and are partially mitigated by increased complexity in other parts of the testing chain, when 5% or less scenarios are transient and geographically variable, especially these days. For how long is pooling actually feasible and how does the lab switch seamlessly from pooling to not pooling? Or pooling for some samples but not for others. Can you make more of an argument about how pooling logistics fit into the overall strategy for surveillance and monitoring in low prevalence populations? Instead of just waiting for pooling to be useless, as it is right now? I realize this may be a bit out of your scope, but could add to your perspective on “the pandemic context”.

Re.1: We are grateful to reviewer for his very interesting analysis and remark.

It is true that the laboratory needs a greater organization that allows the assembly of the pools with perfect traceability. But once the personnel is trained, it does not imply a great extra effort. Regarding when to use the pool strategy and when it does not, it depends on the previous logistics, the epidemiological scenario (the viral prevalence in a given region and moment), and the local health public policy, which is dynamic and varies regarding viral circulation. In other words, an attempt should be made to infer the prevalence percentage of viral infections using recent data from previous tests reported in the area from which the samples come. Another way could be to randomly analyze some individual samples that are representative of the population tested in a given testing center or region. In this way, a percentage of positivity could be estimated and thus decide if the rest of the samples of that population will be analyzed by pooling strategy. Although this could generate a delay in the results.

2-Omicron seems to have changed the game entirely. Although outside the timeframe in which your study was conducted, some recognition of how the pooling strategies you recommend are affected by these developments would be welcome. Any predictions about post-Omicrom scenarios – e.g. specific groups of people benefitting from a daily pooled sample prior to accessing a closed facility.

Re.2: As the reviewer’s stated, the Omicron SARS-CoV-2 variant emerged after the time of this study. However, following your indication, we also add the following paragraph to address your remark at the Discussion section of MS: “The emergence of the Omicron variant, which is more contagious than the previous ones SARS-CoV-2 strains, increases the importance of analyze closed communities through periodic testing. For this, rRT-PCR testing using the pooling strategy has the advantage of being more specific and sensitive than rapid antigen tests, making possible to detect false-negative and asymptomatic people without increasing costs as much, thus contributing to a better management of pandemic in particular time periods.” (line 196-201). 

3-Data processing: I am surprised at all the scripts and databases required to handle two variables (testee identity and Ct values), but I have never worked with such a big sample set! Is there any chance of seeing some of the scripts used, database summaries and visualizations mentioned in this section?

Re.3: The following web link: http://covid.ceprocor.com/ , hosts the site created for the storage and management of data related to pools, which is mentioned in the "Data processing" section. This site is in Spanish and in the home page you can access the links: Results (Resultados), Searches (Busquedas), and Database Information (Info de la base de datos). In Results, data are organized by date (in calendar format). Accessing to a specific day between August 2020 and April 2021, it is possible to find out the results of pooled samples and also individual samples, as they are specified. If a specific pool had a result other than negative, it was disassembled to test each individual sample included in the pool. In this situation, the results of the individual samples can be viewed by choosing from the bottom left section of the page for that day, by internal pool code, and clicking on the button: "Choose internal pool code". All results for the pool samples involved will be seen. The Searches and Database Information sections were used for a better follow-up of the daily work.

All data, for the purposes of this publication, was extracted from the database, accessed directly via MySQL."

4-Minor comments:

4.1 I think most journals would require a comma thousands separator.

Re.4.1: Right, in this revised version all numbers are with comma thousands separator format.

4.2 line 95: “waives” is misspelled.

Re.4.2: This mistake was fixed, now at line 96 says: “The Government of the Province of Córdoba waives the ethical review…”

4.3 line 112: “Beijing” typo.

Re.4.3: This mistake was fixed, now at line 115 says: “TransGene, Beijing, China…”

4.4 line 127-9: To be more concise, I would delete everything from “...exported” to “and interpreted with...”

Re.4.4: We agree, this part was deleted according to the reviewer’s observation, and the paragraph at lines 130-131 now says: “Results obtained by Real Time PCR (sample code and their respective Ct) were organized and converted into comma delimited files.”

4.5 line 139: Misplaced comma.

Re.4.5: This mistake was fixed

4.6 line 144: “All together” instead of “Altogether”

Re.4.6: Now at line 145 says: “In total, 290 pools were disassembled”

4.7 line 145-7: “saved” instead of “allowed to save”. Isn’t “67.5% fewer” the same thing as “3 times reduction” (“testing materials and costs were reduced by two thirds.”)?

Re.4.7: We acknowledge for this observation, now at lines 146-148 says: “As a result, the pooling strategy saved 7,806 tests, that is, 67.5% fewer tests were required for the screening, leading to a two-thirds reduction in costs.”

4.8 line 173-4: “regardless of whether any pool contained positive samples.”

Re.4.8: We agree, now the sentence “regardless of whether any pool contained positive samples” is at lines 174-175. 

4.9 line 179: “disassembly” instead of “deconvolution” – stick to one word for testing all samples in a pool.

Re.4.9: We agree, now the caption of fig 4. at line 180 says “…and the individual positive samples after disassembly…”

4.10 line 191: just “save” instead of “save costs of”.

Re.4.10: Done, now at line 191 says “In addition, it contributes to saving molecular diagnosis…”

4.11 line 195: “those” instead of “persons”, “at greater risk” instead of “risk factors”.

Re.4.11: Done, now at line 195-196 says “…being populated by those who are generally in close contact and/or at greater risk”

4.12 line 196: “optimal pool size” instead of “optimum size pool”

Re.4.12: Done, now at line 202 says “It is important to highlight that optimal pool size must be determined…”

4.13 line 203: “especially for samples” instead of “affecting especially detection of samples”

Re.4.13: Done, now at line 209 says “especially for samples with low viral loads”

4.14 line 212: “the increased sensitivity” instead “a more sensitivity”, “in contrast with the” instead of “contrasting with”

Re.4.14: This change has been done, now at lines 218-219 says: “which could be due to the increased sensitivity of this gene, in contrast with the ORF1ab gene”

4.15 line 216: “disassembling” instead of “disarming”

Re.4.15: This change has been included, now at line 222 says “…to avoid disassembling groups composed of…”

4.16 line 219. Just “labor” instead of “decreasing personnel labor demanding”.

Re.4.16: This change has been done, now at line 225 says “thus, saving 67.5% of costs and labor”

4.17 line 221: “deals” instead of “allows to deal”, “the pandemic context” instead of “a context of pandemics”

Re.4.17: Done, now at line 226 says “The pooling approach addresses a variety of difficulties associated the pandemic context, most notably …”

4.18 line 236: “expanded”, instead of “allow expanding” – replace period with comma after “capabilities”

Re.4.18: Agree, now at lines 239-240 says “The approach of testing in pools was a positive experience that expanded sample processing capabilities, allowing massive testing and early outbreak detections.”

Comments of reviewer #3: 

1-This paper provides a comprehensive study of a pooled testing approach applied in a practical pandemic scenario. To my knowledge, there is not much data available regarding the trade-off between a potential loss of accuracy and the cost reduction resulting from the reduced number of tests. This paper provides such a comparison, giving an overview of the change of ct-values, the number of correct detections and the savings in terms of test reduction. This information is much needed to design public testing strategies based on pooled PCR tests., especially given the fact that antigen tests do not work well for the omicron variant and that there is still some hesitation with regards to the accuracy of pooled tests. For this reason, I find this study quite timely and recommend publication. The paper is also quite well-written, I do not have any specific suggestions or comments.

Re.1: We are grateful to the reviewer for his/her positive comments, recommending the publication of our work.

---

## [Decision Letter · Decision Letter 1]

6 Mar 2022

PONE-D-21-40099R1SARS-CoV-2 detection in multi-sample pools in a real pandemic scenario: a screening strategy of choice for active surveillancePLOS ONE

Dear Dr. Castellaro,

Thank you for submitting your manuscript to PLOS ONE. After careful consideration, we feel that it has merit but does not fully meet PLOS ONE’s publication criteria as it currently stands. Therefore, we invite you to submit a revised version of the manuscript that addresses the points raised during the review process.

ACADEMIC EDITOR: In addition to the comments raised by reviewer # 1, please amend the following:Check the first paragraph of the introduction as there is no references. Is it your own sentences?Replace “in order to” with “to” throughout the textCheck spaces and punctuation throughout the textP letter for statistical value should be uppercase-italic faceFigs captions should not be incorporated in the text. They should be at the bottom of the corresponding oneAvoid using “we, our”. Use impersonal phrasing throughout the textStudy strengths and limitations should be in a separate section, headed as addressed. It should be ahead of the conclusion. Start with the strength of the study, followed by limitations.The conclusion should be in a separate section. What are the clinical relevance and future perspective? Add this to the conclusion sectionProofread the text for grammar and syntax errors ==============================

We look forward to receiving your revised manuscript.

Kind regards,

A. M. Abd El-Aty

Academic Editor

PLOS ONE

Journal Requirements:

Reviewers' comments:

Reviewer's Responses to Questions

**Comments to the Author**

1. If the authors have adequately addressed your comments raised in a previous round of review and you feel that this manuscript is now acceptable for publication, you may indicate that here to bypass the “Comments to the Author” section, enter your conflict of interest statement in the “Confidential to Editor” section, and submit your "Accept" recommendation.

Reviewer #1: All comments have been addressed

Reviewer #2: All comments have been addressed

2. Is the manuscript technically sound, and do the data support the conclusions?

Reviewer #1: Yes

Reviewer #2: Yes

3. Has the statistical analysis been performed appropriately and rigorously? 

Reviewer #1: Yes

Reviewer #2: Yes

4. Have the authors made all data underlying the findings in their manuscript fully available?

Reviewer #1: Yes

Reviewer #2: Yes

5. Is the manuscript presented in an intelligible fashion and written in standard English?

Reviewer #1: Yes

Reviewer #2: Yes

6. Review Comments to the Author

Reviewer #1: Thank you for the clarifications.

Line 108 currently states: "These oropharyngeal swab samples were pooled together into 5 samples based on the order number, without mixing samples from different institutions."

This wouild be more clearly written as "Each group of 5 sequentially obtained samples was pooled, without mixing samples from different institutions."

Reviewer #2: (No Response)

7. PLOS authors have the option to publish the peer review history of their article (what does this mean?). If published, this will include your full peer review and any attached files.

Reviewer #1: No

Reviewer #2: **Yes: **John J. Schellenberg

---

## [Author Response · Author response to Decision Letter 1]

15 Mar 2022

All the authors and I wish to take this opportunity to acknowledge the editor and the reviewers for their constructive comments and valuable recommendations. We have revised the manuscript on our R1 version according to the concerns, and have resubmitted it online. Our responses are listed below.

ACADEMIC EDITOR

1. Check the first paragraph of the introduction as there is no references. Is it your own sentences?

Re. 1- This paragraph was written based on our own experience and the situation that many countries in the world went through. The corresponding citations supporting these sayings have been added.

2. Replace “in order to” with “to” throughout the text

Re. 2- In line 90 of manuscript, “in order to” was replaced by “to”.

3. Check spaces and punctuation throughout the text

Re. 3- Following the editor's comment, spaces and punctuation were checked through the manuscript.

4. P letter for statistical value should be uppercase-italic face

Re. 4- Now, the P letter for statistical value now have the indicated format.

5. Figs captions should not be incorporated in the text. They should be at the bottom of the corresponding one.

Re. 5- At the editor’s request, figure captions were moved from the body text to the bottom of the manuscript.

6. Avoid using “we, our”. Use impersonal phrasing throughout the text

Re. 6- As suggested, most of the personal phrases in the manuscript were changed to impersonal phrases.

7. Study strengths and limitations should be in a separate section, headed as addressed. It should be ahead of the conclusion. Start with the strength of the study, followed by limitations.

Re. 7- The discussion section has been reordered according to the editor’s suggestion. Now, the first part exposes the strengths of the pooling strategy followed by the limitations, both under the corresponding subtitle.

8. The conclusion should be in a separate section. What are the clinical relevance and future perspective? Add this to the conclusion section.

Re. 8- Now the conclusion is in a separate section after the discussion.

9. Proofread the text for grammar and syntax errors

Re. 9- Based on the editor’s comment, English, grammar, and syntax were checked throughout the manuscript.

Reviewer #1: 

10. According to the reviewer’s comment: Line 108 currently states: "These oropharyngeal swab samples were pooled together into 5 samples based on the order number, without mixing samples from different institutions."

This would be more clearly written as "Each group of 5 sequentially obtained samples was pooled, without mixing samples from different institutions."

Re. 10- We thank the reviewer’s advice and following it, the paragraph was changed according to the suggestion (line 102).

---

## [Decision Letter · Decision Letter 2]

21 Mar 2022

SARS-CoV-2 detection in multi-sample pools in a real pandemic scenario: a screening strategy of choice for active surveillance

PONE-D-21-40099R2

Dear Dr. Castellaro,

We’re pleased to inform you that your manuscript has been judged scientifically suitable for publication and will be formally accepted for publication once it meets all outstanding technical requirements.

Kind regards,

A. M. Abd El-Aty

Academic Editor

PLOS ONE

Additional Editor Comments (optional):

Reviewers' comments:

Reviewer's Responses to Questions

**Comments to the Author**

1. If the authors have adequately addressed your comments raised in a previous round of review and you feel that this manuscript is now acceptable for publication, you may indicate that here to bypass the “Comments to the Author” section, enter your conflict of interest statement in the “Confidential to Editor” section, and submit your "Accept" recommendation.

Reviewer #1: All comments have been addressed

2. Is the manuscript technically sound, and do the data support the conclusions?

Reviewer #1: (No Response)

3. Has the statistical analysis been performed appropriately and rigorously? 

Reviewer #1: (No Response)

4. Have the authors made all data underlying the findings in their manuscript fully available?

Reviewer #1: (No Response)

5. Is the manuscript presented in an intelligible fashion and written in standard English?

Reviewer #1: (No Response)

6. Review Comments to the Author

Reviewer #1: (No Response)

7. PLOS authors have the option to publish the peer review history of their article (what does this mean?). If published, this will include your full peer review and any attached files.

Reviewer #1: No

---

## [Editor Report · Acceptance letter]

25 Mar 2022

PONE-D-21-40099R2 

SARS-CoV-2 detection in multi-sample pools in a real pandemic scenario: a screening strategy of choice for active surveillance 

Dear Dr. Castellaro:

I'm pleased to inform you that your manuscript has been deemed suitable for publication in PLOS ONE. Congratulations! Your manuscript is now with our production department. 

Kind regards, 

on behalf of

Prof. A. M. Abd El-Aty 

Academic Editor

PLOS ONE